# Effective Supply Chain Strategies in Addressing Demand and Supply Uncertainty: A Case Study of Ethiopian Pharmaceutical Supply Services

**DOI:** 10.3390/pharmacy12050132

**Published:** 2024-08-28

**Authors:** Arebu Issa Bilal, Umit Sezer Bititci, Teferi Gedif Fenta

**Affiliations:** 1Department of Pharmaceutics and Social Pharmacy, School of Pharmacy, College of Health Sciences, Addis Ababa University, Addis Ababa P.O. Box 9086, Ethiopia; 2Edinburgh Business School, Heriot Watt University, Edinburgh EH14 4AS, UK

**Keywords:** demand, pharmaceutical supply chain, proactive, reactive, strategies, supply

## Abstract

Background: Ensuring the consistent availability of essential medicines is crucial for effective healthcare systems. However, Ethiopian public health facilities have faced frequent stockouts of crucial medications, highlighting systemic challenges such as inadequate forecasting, prolonged procurement processes, a disjointed distribution system, suboptimal data quality, and a shortage of trained professionals. This study focuses on the Ethiopian Pharmaceutical Supply Services (EPSS), known for its highly unstable and volatile supply chain, aiming to identify risks and mitigation strategies. Methods: Using a mixed-method approach involving surveys and interviews, the research investigates successful and less successful strategies, key success factors, and barriers related to pharmaceutical shortages. Results: Proactive measures such as communication, stock assessment, supervision, and streamlined procurement are emphasized as vital in mitigating disruptions, while reactive strategies like safety stock may lack long-term efficacy. The study highlights the importance of aligning supply chain strategies with product uncertainties, fostering collaboration, and employing flexible designs for resilience. Managerial implications stress the need for responsive structures that integrate data quality, technology, and visibility. Conclusions: This study contributes by exploring proactive and reactive strategies, elucidating key success factors for overcoming shortages in countries with unstable supply chains, and offering actionable steps for enhancing supply chain resilience. Embracing uncertainty and implementing proactive measures can help navigate volatile environments, thereby enhancing competitiveness and sustainability.

## 1. Introduction

Universal access to essential medicines is crucial for effective healthcare systems, yet equitable availability remains a global challenge [1,2]. Many nations, particularly in Africa and Asia, struggle with access due to unaffordability and supply chain inefficiencies [3,4]. Drug shortages also impact developed countries like the U.S. and Europe, affecting healthcare quality and patient care [5,6]. These shortages, caused by supply chain disruptions, have financial and patient care consequences [5,7,8].

Factors such as globalization, manufacturing issues, natural disasters, pandemics, and regulatory requirements contribute to medicine shortages [8,9]. Supply chain complexities introduce additional challenges like demand variations and resource limitations [10]. Recent disruptions, including material cost increases [11], demand variations due to the COVID-19 pandemic [12], and supply chain interruptions, have exposed the vulnerability of supply chains [13,14,15,16]. These challenges necessitate a reevaluation of supply chain strategies to enhance resilience and adaptability [17,18].

Inaccurate demand forecasting increases healthcare costs and inefficiencies, affecting drug therapy and patient safety [19,20,21,22,23,24]. A robust supply chain strategy integrates suppliers, manufacturers, and distribution mechanisms to optimize resources and service levels [24]. Addressing shortages, especially in low-income countries, involves managing stock use restrictions, efficient redistribution, and enhancing stakeholder communication [25]. Persistent research gaps in low- and middle-income countries highlight the need for collaborative efforts to tackle this challenge.

An efficient supply chain is integral to organizational success, particularly in healthcare, consumer goods, and agriculture, where shortages can have costly consequences [26,27]. Managing supply and demand uncertainty remains a persistent challenge [28]. Fisher (1997) categorized supply chains into efficient and responsive types based on demand predictability, while Lee (2002) extended this framework to include supply uncertainties. Effective management of these uncertainties involves mechanisms to cope with them and meet customer demands efficiently [29,30].

As supply chain management becomes increasingly complex due to changing customer demands, disruptions, and natural disasters, recent global economic turbulence emphasizes the urgency for resilient and sustainable strategies. Demand uncertainty, arising from factors like seasonal fluctuations and economic conditions, necessitates a flexible supply chain. Supply uncertainty poses challenges such as supplier unreliability, communication issues, and quality concerns. Effective management requires robust supplier relationships, rapid onboarding, and harmonized communication. Operational uncertainties, including transportation delays and production disruptions, impact the supply chain, requiring real-time tracking and effective inventory management. The COVID-19 pandemic and geopolitical events, such as Brexit and conflicts in Ukraine, Gaza, and Yemen, have amplified global supply chain disruptions. Issues like chip shortages and shipping delays underscore the need for strategic supply chain planning and resilience. Geopolitical events disrupt international supply chains, highlighting their vulnerability. The disruption emphasizes the challenges of internal supply chain volatility, necessitating strategies to transform volatility into opportunities.

In Ethiopia, the pharmaceutical supply chain has evolved with EPSS and the Pharmaceutical Logistics Master Plan but still faces issues like stock-outs and wastage [31,32,33]. Comprehensive research is needed to develop effective mitigation strategies, especially in resource-limited settings. This paper aims to identify risks in the EPSS supply chain and strategies to mitigate drug shortages, using literature reviews, surveys, and interviews to guide effective supply chain strategies.

## 2. Literature Review

Our work relates to several streams of literature, including relevant studies that focus on pharmaceutical drug shortages, supply chain risks, factors that contribute to drug product shortages, challenges in the Ethiopian Pharmaceutical Supply Services, and supply chain strategies during demand uncertainties in the pharmaceutical industry.

### 2.1. Drug Shortages Factors That Contribute to Drug Shortages

Drug shortages occur when the total supply of all clinically interchangeable versions of a regulated drug falls short of demand [34]. Despite increased research, reliable data to assess and combat medicine shortages globally remain insufficient [35,36]. In the U.S., medicine shortages have been tracked since the early 2000s but still persist [35,37]. The global rise in essential medicine shortages poses a significant challenge [35,38,39]. Unpredictable demand, limited planning resources, and organizational constraints exacerbate this issue [40]. Drug shortages significantly impact global healthcare, driven by multifaceted factors. These include non-traditional economics where patients lack control over prescriptions and limited transparency in manufacturing [41,42]. Medical supply chains face access, security, and traceability issues [40]. Manufacturing delays, capacity limits, and business decisions disrupt supply, alongside shortages in active pharmaceutical ingredients (APIs) and raw materials. Inefficient inventory practices and restricted distribution exacerbate the issue [39,43,44]. Additional challenges like single sourcing, global sourcing risks, minimal buffer stocks, and a focus on cost reduction increase vulnerabilities [45,46]. Supply chain risks from environmental, organizational, or network sources lead to chaos, inertia, and a lack of confidence [47,48,49]. Regulatory demands and market entry barriers further complicate the situation [35,44]. Understanding these factors is crucial for developing strategies to mitigate the impact of drug shortages on healthcare.

In Ethiopia, frequent stockouts of essential medicines worsen healthcare challenges [31,50]. Effective solutions are urgently needed to address these shortages. Understanding the definition, prevalence, and consequences of drug shortages is crucial for mitigating their impact on healthcare systems and patient well-being.

### 2.2. Supply Chain Risks

Organizations form interconnected networks, increasing interdependence and associated risks [51]. These complex supply chains span globally, with risks multiplying across components and processes [52,53]. Modern technology helps mitigate these risks through digitalization and data analytics [54]. Supply chain risks are categorized into demand risks, such as seasonal fluctuations, and supply risks, like manufacturing challenges [55]. Risks include delays, disruptions, forecast inaccuracies, and intellectual property breaches [56]. External risks stem from events like natural disasters, while internal risks arise from supply chain management [46]. Supply disruptions can lead to customer dissatisfaction, revenue loss, and safety concerns [45]. Understanding and mitigating these risks is crucial for maintaining a robust supply chain.

### 2.3. Challenges in the Ethiopian Pharmaceutical Supply Services

The Ethiopian Pharmaceutical Supply Services (EPSS) plays a vital role in providing affordable and reliable pharmaceuticals to public health facilities in Ethiopia. However, several challenges have hampered its ability to effectively manage drug shortages and ensure the availability of essential medications. One significant challenge faced by the EPSS is inaccurate drug forecasting. A study conducted within the EPSS [57] revealed that the mean percentage forecast error for program commodities stood at 27.8%. This error rate exceeds the acceptable threshold, which is 25% or less [58], and there are variations in forecast accuracy among individual items. For instance, forecast errors for HIV/AIDS and malaria commodities reached 31.71% and 37.25%, respectively, deviating from the expected range [57,59]. The EPSS encounters lengthy procurement lead times, with an average of 137.3 days and orders sometimes experiencing delays of up to 294 days. Such delays significantly impact drug availability. In addition, poor data quality and inaccurate and incomplete consumption reports from service delivery points have led to unacceptable forecasting errors. Coupled with communication and coordination challenges within the EPSS, these further exacerbate supply chain problems [60]. A study involving 40 health facilities in Ethiopia identified human resources, financial resources, infrastructure, and information technology challenges as key issues in pharmaceutical supply chain management [40].

The EPSS grapples with systemic weaknesses, including the absence of a robust system to manage its operations, weak demand planning, extended procurement lead times, inadequate medical equipment management, and a lack of emphasis on market shaping [19,61]. A SWOT analysis conducted in the EPSS unveiled several areas of concern, such as inadequate customer and stakeholder satisfaction, unsatisfactory supplier relationship management, inventory management challenges, weak distribution planning, and a deficient track and trace system. Furthermore, issues like the absence of a robust information system, inadequate financial sustainability planning, cash shortages, untrained staff, inefficient workforce utilization, and organizational culture problems have been identified [61].

In conclusion, the Ethiopian Pharmaceutical Supply Services faces multifaceted challenges that hinder its ability to effectively manage drug shortages and ensure drug availability. Addressing these challenges requires a comprehensive and coordinated effort to improve forecasting, streamline procurement processes, enhance data quality, and bolster the overall pharmaceutical supply chain management system.

### 2.4. Supply Chain Strategies

The pharmaceutical industry involves a range of entities including research-based multinationals, generic manufacturers, biotech startups, and specialized logistics providers [62]. Managing this complex supply chain is challenging due to factors like product diversity, short life cycles, outsourcing, technological advancements, and globalization [30]. Additionally, high costs, prolonged clinical trials, and uncertainties in demand and capacity planning further complicate the industry [63]. In low-income countries, ensuring the availability and affordability of essential medicines is crucial. Strategies such as limiting stock use, transparent redistribution, and proactive shortage management are employed [25]. Effective communication among regulatory bodies, harmonization, and robust reporting systems are essential for mitigating drug shortages [37,64]. Investments in research, policies, guidelines, education, and training are also necessary [25,38].

Supply chain strategies must account for demand and supply uncertainties. Fisher’s framework categorizes products as functional or innovative based on demand predictability and supply processes as stable or evolving [29]. For stable products, efficient supply chains focus on productivity and logistics optimization [30]. Innovative products require responsive strategies to adapt to changing demands [65]. Companies dealing with both innovative products and unstable supply processes must use a combination of risk-hedging and responsive strategies to manage uncertainties [30]. Global collaboration, particularly among low-income countries, is essential for developing effective mitigation strategies for drug shortages. International regulatory bodies should work together on a unified definition and global mitigation plan. Nationally, countries should establish proactive systems for drug shortage notification, reporting, and tracking. Policies should promote robust supply chains, encourage quality manufacturing systems, and prioritize the production of high-risk medicines. Training healthcare professionals and educating the public is also critical to minimize health-related consequences [25].

In conclusion, addressing demand uncertainties in the pharmaceutical industry requires a nuanced understanding of product characteristics, supply processes, and effective supply chain strategies. Also, it is clear that collaboration at both national and international levels is critical to mitigating the global issue of drug shortages across all economic levels (Table 1).

The aim of this study is to understand the risks in the EPSS supply chains and identify strategies that could be engaged to mitigate the risk of drug shortages. Thus, the specific research questions of this paper are:Which supply chain strategies, individually or in combination, provide an effective means for minimizing the drug shortages in Ethiopian Pharmaceutical Supply Services?What strategies have been the least effective in reducing or eliminating supply chain disruptions?What are the barriers that prohibit supply chain management strategies from becoming successful?

## 3. Research Method

### 3.1. Context

The Ethiopian Pharmaceutical Supply Services (EPSS) falls under the Ministry of Health, which is responsible for the forecasting, procurement, warehousing, and distribution of pharmaceuticals throughout the country. It has 19 branches, which are found in all regions of Ethiopia. The EPSS operates as a government agency under the Ministry of Health, tasked with centralizing the procurement and distribution of pharmaceuticals and medical supplies for the public health sector. It plays a crucial role in ensuring the availability and accessibility of essential medicines and medical products across the country.

The EPSS is entrusted with procuring pharmaceutical products and medical supplies through competitive bidding processes, aiming to secure high-quality products at reasonable prices. It maintains central medical stores strategically positioned throughout the country, acting as warehouses that manage and distribute the procured pharmaceuticals to regional and district health facilities. The EPSS places significant emphasis on improving supply chain management practices to reduce stock outs and minimize the wastage of pharmaceuticals. Through the use of information systems and logistics management tools, the agency strives to optimize inventory and enhance the efficiency of the distribution process.

Furthermore, the EPSS collaborates with the Ethiopian Food and Drug Authority (EFDA) to ensure that all procured medicines and medical supplies meet the required quality standards. The agency actively works towards enhancing access to essential medicines, particularly in underserved and remote areas, through its distribution networks and partnerships with health authorities and stakeholders. Additionally, the EPSS engages in collaboration with international organizations, donors, and development partners to strengthen its capacity and continually improve pharmaceutical supply chain practices.

### 3.2. Research Design

This study utilized a cross-sectional case study research design to assess the effectiveness of supply chain strategies employed to deal with drug shortages within the EPSS. The research was structured into two sequential parts, a survey and semi-structured interviews.

#### 3.2.1. Part 1—Survey

The first part of the study comprised a survey that included socio demographic questions (such as years of experience in the sector, education level, and professions), as well as questions about supply chain strategies and their effectiveness. In this section, the respondents were presented with different supply chain strategies and asked to rate their effectiveness using a 5-point Likert scale, ranging from “highly effective” to “never effective”. The questionnaire, which is provided in Appendix A, was completed by 50 respondents with the demographic profile, as summarized in Table 2.

After the data collection phase, the questionnaires were coded, and the completeness of the data was thoroughly checked to ensure accuracy and consistency. Subsequently, the Likert-scale responses were converted into numerical values for analysis using multipliers commonly used in quality function deployment [92,93,94]. The use of 9 for highly effective, 3 for effective, 1 for limited effectiveness, and 0 for do not know ensures significant distances between highly effective and less effective strategies. The scores were then summed to prioritize the strategies based on their overall effectiveness (Table 3).

#### 3.2.2. Part 2—Interviews

In the second part of the study, we further narrowed down the participants to 34 and conducted semi-structured interviews with the selected participants. During this stage, in-depth discussions were held to explore the strategies further. The participants were asked about the critical factors they consider when implementing strategies to mitigate or eliminate supply chain disruptions. Additionally, insights were gathered regarding which strategies were perceived as the least effective in addressing supply chain and distribution channel disruptions.

### 3.3. Reflexivity

None of the authors of this study were affiliated with the EPSS. Three team members (A.B., T.G., and U.S.) possess extensive experience and expertise in qualitative research methodology. The interviews were conducted by one of the local authors, A.B. All authors had prior training and a thorough understanding of the study context and values, enabling them to perform transcription and crosscheck the contents during translation. The authors discussed an initial analysis of potential informants and created schedules to conduct the interviews at the participants’ offices.

### 3.4. Selection of Participants for the Interview

Participants were chosen based on their roles, ensuring a comprehensive representation across all departments within EPSS. Their involvement was contingent on obtaining consent, with efforts made to guarantee their full understanding of the study’s objectives and the confidentiality of their privacy. Emphasis was placed on the voluntary nature of participation, and oral consent was obtained from each participant.

Informed by the findings of the survey (Table 3), the interviews focused on surfacing the types of strategies employed to manage supply disruptions in the pharmaceutical supply chain and distribution channels. Additionally, participants were asked about the most crucial factors considered during the implementation of strategies to reduce or eliminate supply chain disruptions, the identification of strategies that have been least effective in addressing disruptions, the specifics of supply chain management strategies used to tackle disruptions, and an exploration of barriers hindering the success of supply chain management strategies.

After obtaining participants’ consent, in-depth interviews were conducted. Both tape recordings and notes were employed for comprehensive documentation. The researchers facilitated the interviews by posing questions naturally, attentively listening to participants’ responses, and incorporating follow-up questions and probes based on those responses. These interviews took place either face-to-face or online, involving one interviewer and one participant. The choice of interview location was convenient for participants, occurring in their offices or the researcher’s office. Online interviews were conducted in participants’ homes based on their preferences and residences. The demographics of interview participants are summarized in Table 4.

### 3.5. Ethical Approval

Ethical approval was obtained from Addis Ababa University’s Institutional Review Board and the School of Pharmacy’s Ethics Review Committee (Protocol 010/22/SoP). The EPSS Directors also granted permission. The participants were assured of their right to withdraw and gave written consent. Privacy was maintained, using pseudonyms for interviewees. Records were securely stored.

### 3.6. Analysis

Data collection and analysis were conducted simultaneously. Interviews were recorded, transcribed, and translated. A line-by-line coding approach yielded 70 codes, which were merged into 9 family codes and then into 4 main themes. Thematic analysis was included to identify and organize central themes. An inductive approach guided further data collection and analysis until theoretical saturation was reached.

## 4. Results

The survey findings are summarized in Table 2. In part 2, semi-structured interviews informed by the survey provided deeper insights into drug shortages and pharmaceutical supply chain management, identifying four themes detailed below (Figure 1).

Themes 1—Most Successful Strategies
Communication and Stock Status Assessment:Regular central-level stock assessments and hub updates optimize planning and prevent stockouts or overstocking. A participant noted, “*Good communication and sharing stock status information with hubs is crucial.*” (Participant 8)Supportive Supervision and Training:Providing guidance and training to staff minimizes drug shortages and motivates them. A participant stated, “*Supportive supervision and training are very helpful for mitigating drug shortages.*” (Participant 23)Streamlining Procurement Processes:Early tender notifications expedite supply acquisition, reduce lead times, and enhance supply chain responsiveness, despite potential challenges. A respondent remarked, “*Early tendering and procurement notification is crucial.*” (Participant 17)Enhancing Supplier Relationship Management:Collaboration with suppliers and stakeholders improves communication and response coordination during disruptions. A participant emphasized, “*Effective supplier relationship management ensures a seamless supply chain.*” (Participant 10)These interconnected strategies—Communication and Stock Status Assessment, Supportive Supervision and Training, Streamlining Procurement Processes, and Enhancing Supplier Relationship Management—demonstrate a holistic and collaborative approach to optimizing pharmaceutical supply chain management.Theme 2—Least Successful Strategies Used by EPSS
Rationing:Rationing controls pharmaceutical supply allocation to ensure fair distribution during scarcity. Though not as proactive as other strategies, it can help balance supply and demand. However, some participants question its effectiveness, with one stating, “*Rationing will not solve the problem; it is just taken as a painkiller.*” (Participant 15)Redistribution:Redistributing products between hubs can incur high costs and time. Efficient redistribution requires strong communication and coordination. A respondent noted, “*Redistribution is costly and time-taking, creating a burden on professionals.*” (Participant 29)Restricted Tenders and Emergency Procurement:Restricted tenders and emergency procurement can meet immediate needs but disrupt routine practices. This may cause future shortages. One participant highlighted, “*Emergency situations make us neglect routine practice, creating another shortage.*” (Participant 34)In summary, while rationing, redistribution, and restricted tenders with emergency procurement are less effective strategies, they can still mitigate shortages temporarily until more proactive methods are implemented. These methods underscore the need for balancing reactive and proactive approaches in pharmaceutical supply chain management.Theme 3—Key Success Factors
Data Quality, Technology, and Automation, Supply Chain Visibility, Demand Forecasting:Accurate data are crucial for effective supply chain management. Real-time, high-quality data enable informed decisions and prompt responses to disruptions. A participant noted, “*Data quality plays a significant role in the whole cycle of supply chain management*” (Participant 28). Technology and automation reduce manual errors and enhance efficiency. Transparent supply chain visibility aids in better planning and risk management. Proper demand forecasting minimizes overstocking and stockouts, ensuring a smooth flow of products.Procurement Lead Time and Supplier Reputation and Relationship Management:Shortening procurement lead time is critical. Delays can affect the entire supply chain. A respondent said, “*Reducing the procurement lead time will have a great impact on the whole supply chain process*” (Participant 26). Reliable suppliers are vital for timely deliveries and quality products. Effective supplier management fosters strong relationships. As one participant highlighted, “*Managing suppliers is quite important*” (Participant 33).Compliance and Regulatory Considerations, Risk Management Strategies:Adhering to industry regulations avoids delays and disruptions. Developing risk management strategies specific to the pharmaceutical supply chain is essential. Identifying risks, assessing their impact, and establishing contingency plans enhance resilience. Considering these factors helps organizations proactively address disruptions, leading to a more robust and reliable pharmaceutical supply chain.Theme 4—Barriers within EPSS for Success
Several barriers impede effective supply chain management at EPSS, highlighting logistical and organizational challenges.
Data Quality Problems: Poor data quality due to inaccurate demand reports from health professionals compromises reliability. A respondent noted, “*Professionals from health facilities often submit inaccurate demand reports, assuming EPSS won’t fulfill their requested amounts, compromising data quality.*” (Participant 22)Shortage of Hard Currency: Currency shortages delay procurement and disrupt the supply chain. “Even with effective forecasting and tender processes, currency shortages in the country delay EPSS’s ability to procure products on time.” (Participant 19)Poor Communication within EPSS and Between Hubs: Internal communication gaps worsen shortages. “*Poor communication within departments and between central EPSS and hubs exacerbates shortages.*” (Participant 11)Failure of Local Manufacturers and Suppliers to Deliver: Local manufacturers, reliant on imported raw materials, often fail to meet commitments. “*Local manufacturers, reliant on global supply chains for raw materials, coupled with low capacity, impact product availability.*” (Participant 9)Bureaucratic Processes and the Bullwhip Effect: Bureaucratic delays and intensified demand fluctuations add complexity.Addressing these challenges requires improving data quality, streamlining procurement, enhancing communication, investing in human resources, and tailoring procurement laws to the pharmaceutical sector. These measures can lead to a more successful and resilient supply chain system, benefiting both the EPSS and the broader pharmaceutical industry.

## 5. Discussion

The primary aim of this research was to comprehend the risks within EPSS supply chains and identify strategies for mitigating the risk of drug shortages. Building on the literature review, the study explored various shortage-mitigating strategies through a survey and participant interviews. The findings highlighted successful and less successful strategies, key success factors, and barriers within the EPSS concerning drug shortages. This study’s contributions are twofold. First, it advances our theoretical understanding by exploring proactive and reactive supply chain strategies for mitigating drug shortages within the pharmaceutical supply chain. Second, it sheds light on key success factors crucial for overcoming drug shortages in resource-limited countries, providing practical insights.

The literature review emphasized the dichotomy of proactive and reactive strategies in supply chain risk management. Proactive strategies involve anticipatory and preventive actions to address potential challenges before they occur, while reactive strategies respond to challenges after they have manifested [95]. These strategies include supplier development, supply chain contracts, contingency planning, and disaster management [96]. Studies from various contexts have identified specific strategies such as establishing information systems, imposing public service obligations, influencing trade rules, and forming committees to collaboratively tackle shortages [97]. The EPSS study aligns with these findings, offering insights into the effectiveness of such strategies within a developing country’s pharmaceutical supply chain (Figure 2).

Additionally, the research provides a unique perspective by delving into specific proactive strategies employed by EPSS, such as communication, stock status assessment, supportive supervision and training, and streamlining procurement processes. Resource sharing through strategic alliances was also identified as a proactive resilience strategy [9]. Reactive strategies discussed in the literature, such as safety stock, capacity buffers, and supplier backups, were found to be used in response to uncertainty in the pharmaceutical supply chain [98]. It is important to understand that both proactive and reactive strategies do not stand alone as single entities; they interact with each other. This interaction is illustrated in Figure 2. For instance, **data quality, technology, and automation** *enable better and more frequent* **communication and stock status assessment,** which in turn *improves* **rationing**, **redistribution**, **emergency management**, **compliance**, and **risk management**. The role of data quality, technology, and automation also helps us to easily communicate and address information.

Furthermore, our findings reveal that supportive supervision and training are crucial when dealing with uncertainty of demand and supply, and this was evident in one of the studies conducted by Bilal et al. [19], which emphasized the pivotal role of training in fostering workforce competence and motivation. While short-term training interventions were deemed crucial for immediate skill enhancement, participants emphasized the long-term imperative of elevating the quality of education programs, including degree-level education, in cultivating a pipeline of skilled professionals.

The study adds depth by examining how the EPSS utilizes these strategies to address medicine shortages. Furthermore, the study acknowledges the scarcity of research on supply chain risk management in developing countries. This gap is critical because of the heightened uncertainty in such regions, emphasizing the study’s relevance to a more fragile economic and political structure.

In terms of managerial implications, the study underscores the importance of a responsive supply chain structure for mitigating risks effectively. It highlights the need for a nuanced approach, understanding that certain strategies may not be effective in isolation and that a combination of proactive and reactive strategies may be necessary. Managers are encouraged to focus on key success factors, including data quality, technology, automation, and supply chain visibility while addressing barriers like data quality problems and communication issues. The success of reactive and proactive strategies hinges on their alignment with the company’s overall supply chain goals and the particular uncertainties encountered. Combining both strategies in a balanced manner can create a more resilient supply chain that effectively handles uncertainties. In essence, although reactive and proactive strategies have distinct roles, their interaction is vital for optimal supply chain flexibility. Companies that effectively merge these approaches can boost their responsiveness to uncertainty and advance long-term enhancements in their supply chain design and operations.

While interpreting the findings, it is essential to acknowledge the study’s limitations, including its focus solely on EPSS. The reliance on self-reported data and a relatively small sample size may limit the generalizability of insights. Future research should adopt broader sampling methods to enhance the study’s external validity. In conclusion, despite these limitations, this study provides valuable insights into pharmaceutical supply chain management within EPSS. The identified strategies, success factors, and barriers offer actionable steps for stakeholders to enhance the resilience and efficiency of pharmaceutical supply chains in Ethiopia. The continuous evaluation and adaptation of these strategies are imperative for maintaining a responsive and sustainable pharmaceutical supply chain in the country.

## 6. Conclusions

This paper has shed light on the proactive and reactive strategies that are important in reducing drug shortages in the Ethiopian pharmaceutical supply chain. By showing how these strategies work together, we emphasize the need to use both for effective mitigation. The distinction between effective and ineffective strategies in this study is a useful guide for supply chain management practitioners. It gives clear insights into areas that need improvement and changes in their current practices. This practical advice is crucial for improving supply chain operations and building resilience against drug shortages.

In conclusion, this paper not only contributes to the existing knowledge by unraveling the dynamics of mitigating drug shortages but also provides a roadmap for future exploration. The intricacies of this study pave the way for further research, encouraging a deeper understanding of how to address and mitigate drug shortages in the complex landscape of resource-limited countries.

## Figures and Tables

**Figure 1 pharmacy-12-00132-f001:**
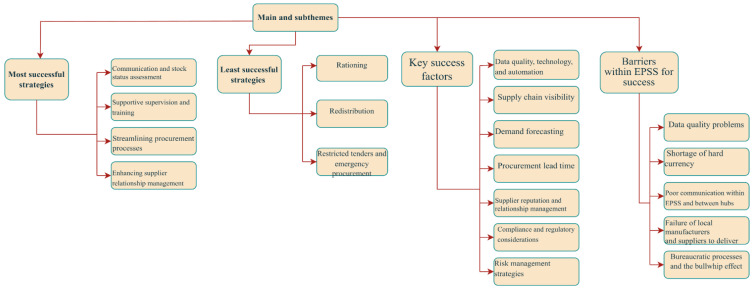
The main themes and sub-themes of the thematic analysis.

**Figure 2 pharmacy-12-00132-f002:**
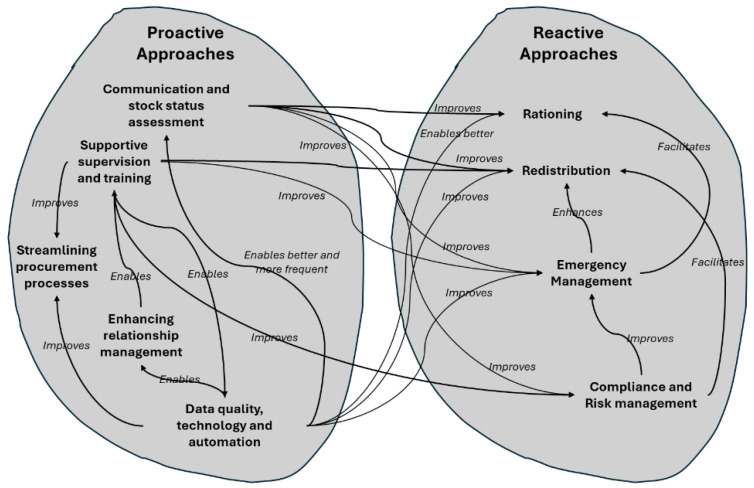
How the different strategies interact to aid improved management of supply chain performance under demand uncertainties.

**Table 1 pharmacy-12-00132-t001:** Supply chain strategies for managing uncertainties.

Supply Chain Strategies for Management of Drug Shortages	References
Using different forecasting methods	[66]
Market monitoring enhancement	[67]
Selecting drug suppliers with short lead times	[68]
Strengthen demand forecasting	[69,70]
Improve inventory management along supply chains	[71]
Implement buffer stock and emergency procurement mechanisms	[37]
Increasing the quantity of the safety stock	[72]
Establishing feedback loops among stakeholders involved in the forecasting process	[42]
Shortage reporting and tracking system	[25,67,73,74,75,76,77]
Maintain end-to-end public health supply chain visibility	[67]
Engage in systematic de-bottlenecking	[78]
Minimizing wastage of products	[25,76,79]
Strengthening access to medicines through public-private partnerships	[80]
Increase communication among stakeholders	[34,41,64,67,70,71,74,80,81]
Advance notification system when there is shortage	[67,74,82]
Improving the procurement lead time	[74]
Restriction and quota for those pharmaceuticals with unstable demand	[25]
Education and training for the staff on how to handle shortages	[25,70,74]
Establishing robust supply chain management systems	[83]
Fast decision-making	[42]
Collaborative planning, forecasting and replenishment (CPFR)	[84,85]
Developing guidelines on procurement strategies under a crisis	[86,87]
The Reorder Level Strategy, also known as a two-bin system	[88]
Communicating with the purchasing officer and provide what you have in the stock	[25,89]
Time Postponement	[90]
Buy to Order (BTO) (also known as Purchase-to-Order)	[91]
Redistribution of the stock	[25]

**Table 2 pharmacy-12-00132-t002:** Socio demographic profile of respondents in EPSS in 2023.

Socio Demographic Variables	Frequency	Percent
Gender		
Female	16	32
Male	34	68
Age in years		
25–30	16	32
30–35	19	38
Year of experience		
1–5	16	32
Greater than 5	34	68
Profession		
Pharmacist	37	74
Laboratory	8	16
Biomedical engineer	5	10
Level of education		
Degree	31	62
MSc	19	38
Current working department		
Contract management	6	12
Quantification market shaping	13	26
Tender management	12	24
Warehouse inventory management	19	38

**Table 3 pharmacy-12-00132-t003:** Summary of findings from the survey of supply chain practitioners.

Supply Chain Strategies for Management of Drug Shortages	Highly Effective (x9)	Effective (x3)	Don’t Know (x0)	Less Effective (x − 3)	It Will Never Be Effective (x − 9)	Weighted Score	Rank
Using different forecasting methods	28 (56%)	15 (30%)	5 (10%)	2 (4%)	0	291	1
Market monitoring enhancement	27 (54%)	17 (34%)	5 (10%)	1 (2%)	0	291	1
Selecting drug suppliers with short lead times	19 (38%)	21 (42%)	7 (14%)	3 (6%)	0	225	2
Strengthen demand forecasting	20 (40%)	17 (34%)	8 (16%)	5 (10%)	0	216	3
Improve inventory management along supply chains	19 (38%)	20 (40%)	6 (12%)	5 (10%)	0	216	3
Implement buffer stock and emergency procurement mechanisms	15 (30%)	26 (52%)	9 (18%)	0	0	213	4
Increasing the quantity of the safety stock	15 (30%)	26 (52%)	6 (12%)	3 (6%)	0	204	5
Establishing feedback loops among stakeholders involved in the forecasting process.	16 (32%)	23 (46%)	6 (12%)	4 (8%)	0	201	6
Shortage reporting and tracking system	15 (30%)	24 (48%)	8 (16%)	3 (6%)	0	198	7
Maintain end-to-end public health supply chain visibility	16 (32%)	21 (42%)	10 (20%)	3 (6%)	0	198	7
Engage in systematic de-bottlenecking	10 (20%)	36 (72%)	4 (8%)	0	0	198	7
Minimizing wastage of products	14 (28%)	26 (52%)	7 (14%)	3 (6%)	0	195	8
Strengthening access to medicines through public-private partnerships	15 (30%)	24 (48%)	4 (8%)	4 (8%)	0	195	8
Increase communication among stakeholders	17 (34%)	18 (36%)	9 (18%)	6 (12%)	0	189	9
Advance notification system when there is shortage	16 (32%)	21 (42%)	6 (12%)	6 (12%)	0	189	9
Improving the procurement lead time	18 (36%)	16 (32%)	12 (24%)	2 (4%)	2 (4%)	186	10
Restriction and quota for those pharmaceuticals with unstable demand	15 (30%)	19 (38%)	13 (26%)	3 (6%)		183	11
Education and training for the staff on how to handle when there is shortage	12 (24%)	29 (58%)	4 (8%)	4 (8%)	0	183	11
Establishing robust supply chain management systems	13 (26%)	23 (46%)	9 (18%)	3 (6%)	0	177	12
Fast decision-making	12 (24%)	26 (52%)	6 (12%)	4 (8%)	0	174	13
Collaborative planning, forecasting and replenishment (CPFR)	12 (24%)	26 (52%)	6 (12%)	6 (12%)	0	168	14
Developing detailed guidelines on procurement strategies under a crisis	11 (22%)	29 (58%)	2 (4%)	8 (16%)		162	15
The Reorder Level Strategy, also known as a two-bin system	13 (26%)	22 (44%)	5 (10%)	6 (12%)	1 (2%)	156	16
Communicating with the purchasing officer and provide what you have in the store	16 (32%)	15 (30%)	5 (10%)	9 (18%)	2 (4%)	144	17
Time Postponement	7 (14%)	28 (56%)	11 (22%)	4 (8%)	0	135	18
Buy to Order (BTO) (also known as Purchase-to-Order)	11 (22%)	16 (32%)	15 (30%)	5 (10%)	0	132	19
Redistribution of the stock	13 (26%)	28 (56%)	6 (12%)	3 (6%)	9	111	20

**Table 4 pharmacy-12-00132-t004:** Interview participant profile.

Socio Demographic Variables	Frequency	Percent
Gender		
Female	14	41.17%
Male	20	58.82%
Age in years		
25–30	12	35.29%
30–35	14	41.17%
Greater than 35	8	23.52%
Year of experience		
1–5	16	32%
Greater than 5	34	68%
Profession		
Pharmacist	26	76.47%
Laboratory	4	11.76%
Biomedical engineer	4	11.76%
Level of education		
Degree	22	64.70%
MSc	12	35.30%
Current working department		
Contract management	4	11.76%
Quantification market shaping	10	29.41%
Tender management	8	23.52%
Warehouse inventory management	10	29.41%

## Data Availability

The raw data supporting the conclusions of this article will be made available by the authors upon request.

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
