# Peer review of "Effective Supply Chain Strategies in Addressing Demand and Supply Uncertainty: A Case Study of Ethiopian Pharmaceutical Supply Services"

_pharmacy, 2024, doi:10.3390/pharmacy12050132_

Round 1
Reviewer 1 Report
Comments and Suggestions for Authors
Conversion scale you used in Table 3 is very arbitrary without any valid test. The results may be changed because of this arbitrary assignment. Validity test may provide better picture of this process. Without validity test, it is very risky in assigning he Rank because the ranks may be used (as you implied) as policy directions.
Literature review is well done.
Author Response
Dear Reviewers,
Thank you for the opportunity to revise our manuscript. We greatly appreciate the constructive criticisms and suggestions provided by the reviewers, which have significantly improved the manuscript. We have incorporated the comments from the referees as thoroughly as possible (see our point-by-point response below) and hope the revised proposal is now acceptable.
Regards,
Arebu Issa
Reviewer: # 1
Some specific comments:
Conversion scale you used in Table 3 is very arbitrary without any valid test. The results may be changed because of this arbitrary assignment. Validity test may provide better picture of this process. Without validity test, it is very risky in assigning he Rank because the ranks may be used (as you implied) as policy directions.
- Thank you for your comments and concern we did not assign just any arbitrary number rather we have used quality function deployment(QFD) which is a customer-driven planning process for product development as well as for quality management in general. It is also “a flexible tool that can be fashioned to be effective in a wide range of applications and for many types of organizations” with many commonly known qualitative and quantitative benefits. There are many studies which used this methods and we have indicated the references in the manuscript.
Literature review is well done.
- Thank you

Reviewer 2 Report
Comments and Suggestions for Authors
The study is conducted correctly and the conclusions are in line with the results. Regarding the discussion, it is my opinion that little consideration has been given to staff training processes and it could be implemented. Another useful suggestion that could be taken into consideration is the regional assessment of disease prevalence in order to predict the type and quantity of drugs needed for routine supplies; In this way, space would be left for emergencies. I think overall the authors could implement these topics.
Author Response
Dear Reviewers,
Thank you for the opportunity to revise our manuscript. We greatly appreciate the constructive criticisms and suggestions provided by the reviewers, which have significantly improved the manuscript. We have incorporated the comments from the referees as thoroughly as possible (see our point-by-point response below) and hope the revised proposal is now acceptable.
Regards,
Arebu Issa
Reviewer: # 2
Some specific comments:
The study is conducted correctly and the conclusions are in line with the results. Regarding the discussion, it is my opinion that little consideration has been given to staff training processes and it could be implemented.
- Thank you for your suggestions. We have included and expanded on the ideas related to supportive supervision and training in the discussion section of the manuscript.
Another useful suggestion that could be taken into consideration is the regional assessment of disease prevalence in order to predict the type and quantity of drugs needed for routine supplies; In this way, space would be left for emergencies. I think overall the authors could implement these topics.
- As you clearly indicated, Ethiopia's geography is diverse, with regions that include some of the highest mountains and the lowest points on earth. This variation in geography results in different climates and disease patterns, which in turn affects the types of medications needed in specific regions. However, we chose not to delve into this aspect in detail, as it would broaden the scope of the manuscript and potentially detract from the main focus for readers.

Reviewer 3 Report
Comments and Suggestions for Authors
The manuscript in question provides an interesting insight into the workings of pharmaceutical supply which could be of interest for policy makers in different countries.
I have several suggestions for the respectable authors.
Title:
The word “Title” should be deleted.
First impression:
In the first two to three pages there are countless sentences of the following type: “A, B, C… suggest/indicate/show supply disruptions”. These repetitions are not necessary. Overall, the Introduction can be rephrased and shortened – doing so might help with the repetitions.
There are several doubled spaces throughout the text (e.g. several in Section 2.3, 1st paragraph).
Section 2.3, 1st paragraph
The authors state “A study conducted within EPSS [57] revealed that the mean percentage forecast error for program commodities stood at 27.8%. This error rate exceeds the acceptable threshold,…” it would be helpful to define the acceptable threshold and the appropriate reference.
3. Research method, 3.1 Context
This sentence is missing a verb? “It has 19 branches, which found in all regions of Ethiopia.”
3.2. Part 1
Were all the respondents employees of EPSS? This was never made clear. How many employees are there in EPSS? What is the proportion of surveyed employees (if they are, in fact, EPSS employees)?
Table 3 should be somehow corrected in order for it not to appear so disarranged.
Sincerely,
The reviewer
Author Response
Dear Reviewers,
Thank you for the opportunity to revise our manuscript. We greatly appreciate the constructive criticisms and suggestions provided by the reviewers, which have significantly improved the manuscript. We have incorporated the comments from the referees as thoroughly as possible (see our point-by-point response below) and hope the revised proposal is now acceptable.
Regards,
Arebu Issa
Reviewer: # 3
Some specific comments:
The manuscript in question provides an interesting insight into the workings of pharmaceutical supply which could be of interest for policy makers in different countries.
I have several suggestions for the respectable authors.
Title:
The word “Title” should be deleted.
- Thank you and done accordingly
First impression:
In the first two to three pages there are countless sentences of the following type: “A, B, C… suggest/indicate/show supply disruptions”. These repetitions are not necessary. Overall, the Introduction can be rephrased and shortened – doing so might help with the repetitions.
There are several doubled spaces throughout the text (e.g. several in Section 2.3, 1st paragraph).
- Thank you modified accordingly
Section 2.3, 1st paragraph
The authors state “A study conducted within EPSS [57] revealed that the mean percentage forecast error for program commodities stood at 27.8%. This error rate exceeds the acceptable threshold,…” it would be helpful to define the acceptable threshold and the appropriate reference.
- Thank you the acceptable forecast accuracy by the EPSS is less than 25% we have included this information and referenced it accordingly
- Research method, 3.1 Context
This sentence is missing a verb? “It has 19 branches, which found in all regions of Ethiopia.”
- Thank you. We have added the verb and made the necessary improvements accordingly.
- It has 19 branches, which are found in all regions of Ethiopia
3.2. Part 1
Were all the respondents employees of EPSS? This was never made clear. How many employees are there in EPSS? What is the proportion of surveyed employees (if they are, in fact, EPSS employees)?
- Thank you for your questions. All the respondents are part of EPSS. At the time of data collection, there were 120 pharmacy professionals, comprising 41.6% of the total.
Table 3 should be somehow corrected in order for it not to appear so disarranged.
- Thank you we have modify it accordingly

Reviewer 4 Report
Comments and Suggestions for Authors
REVIEW REPORT FOR THE STUDY “EFFECTIVE SUPPLY CHAIN STRATEGIES IN ADDRESSING DEMAND AND SUPPLY UNCERTAINTY: A CASE STUDY OF ETHIOPIAN PHARMA-CEUTICAL SUPPLY SERVICES”
Journal: Pharmacy
The paper "Effective Supply Chain Strategies in Addressing Demand and Supply Uncertainty: A Case Study of Ethiopian Pharmaceutical Supply Services", performs a study on identifying risks in the Ethiopian Pharmaceutical Supply Services (EPSS) supply chain and strategies to mitigate drug shortages, using literature review, surveys, and interviews to guide effective supply chain strategies.
Title and summary. The title and abstract express well the object of study, objectives, and results of the article.
Structure of the article. The contents are well organized and they adhere to the IMRaD structure. It includes a theoretical framework of the research problem.
Focusing on the opportunity of the study, it must be said that it is useful work given that pharmaceutical shortages, caused by supply chain disruptions, have financial and patient care consequences.
Materials and methods.
Regarding the material and methods section, the methodology is tailored to the object of study and the objectives and is explained in a transparent manner while it has been validly applied to guarantee the results and authors indicate the number of subjects in the sample in the different phases of the study.
Results.
The results are significant and they are presented in an adequate and understandable way through narration but, it would be of interest, if the authors could propose, in a complementary manner, diagrams or conceptual schemes. Visual displays have been widely used in the grounded theory tradition (Strauss, 1999), and some authors argue that analytical diagrams are an integral part of the methodology for developing grounded theory (Charmaz, 2006). It is advisable throughout the coding process to include visual representations of the developing theory in the analytic memos, as diagrams allow for the integration of the coded data.
Likewise, Clarke (2005), in his reworking of the interactionist analytical focus of grounded theory based on postmodernist premises and Foucauldian discourse analysis, proposes the elaboration of diagrams that he calls situational maps. These maps make it possible to understand the social actions and processes under study.
The results justify and relate to the objectives and methods and the results are of sufficient interest.
Discussion.
The discussion appropriately compares the study results with other works, highlighting the main study findings. The 29.98% of the bibliography cited in the study belongs to the previous five years.
Overall, it is an interesting study and should be considered for publication in Pharmacy, once the minor revisions proposed have been resolved.

Author Response
Dear Reviewers,
Thank you for the opportunity to revise our manuscript. We greatly appreciate the constructive criticisms and suggestions provided by the reviewers, which have significantly improved the manuscript. We have incorporated the comments from the referees as thoroughly as possible (see our point-by-point response below) and hope the revised proposal is now acceptable.
Regards,
Arebu Issa
Reviewer: # 4
Some specific comments:
Comments and Suggestions for Authors
REVIEW REPORT FOR THE STUDY “EFFECTIVE SUPPLY CHAIN STRATEGIES IN ADDRESSING DEMAND AND SUPPLY UNCERTAINTY: A CASE STUDY OF ETHIOPIAN PHARMA-CEUTICAL SUPPLY SERVICES”
Journal: Pharmacy
The paper "Effective Supply Chain Strategies in Addressing Demand and Supply Uncertainty: A Case Study of Ethiopian Pharmaceutical Supply Services", performs a study on identifying risks in the Ethiopian Pharmaceutical Supply Services (EPSS) supply chain and strategies to mitigate drug shortages, using literature review, surveys, and interviews to guide effective supply chain strategies.
Title and summary. The title and abstract express well the object of study, objectives, and results of the article.
Structure of the article. The contents are well organized and they adhere to the IMRaD structure. It includes a theoretical framework of the research problem.
- Thank you so much
Focusing on the opportunity of the study, it must be said that it is useful work given that pharmaceutical shortages, caused by supply chain disruptions, have financial and patient care consequences.
- Thank you so much
Materials and methods.
Regarding the material and methods section, the methodology is tailored to the object of study and the objectives and is explained in a transparent manner while it has been validly applied to guarantee the results and authors indicate the number of subjects in the sample in the different phases of the study.
- Thank you so much
Results.
The results are significant and they are presented in an adequate and understandable way through narration but, it would be of interest, if the authors could propose, in a complementary manner, diagrams or conceptual schemes. Visual displays have been widely used in the grounded theory tradition (Strauss, 1999), and some authors argue that analytical diagrams are an integral part of the methodology for developing grounded theory (Charmaz, 2006). It is advisable throughout the coding process to include visual representations of the developing theory in the analytic memos, as diagrams allow for the integration of the coded data.
Likewise, Clarke (2005), in his reworking of the interactionist analytical focus of grounded theory based on postmodernist premises and Foucauldian discourse analysis, proposes the elaboration of diagrams that he calls situational maps. These maps make it possible to understand the social actions and processes under study.
The results justify and relate to the objectives and methods and the results are of sufficient interest.
- Thank you. We have modified the figure and indicated the relationship between the proactive and reactive strategies.
Discussion.
The discussion appropriately compares the study results with other works, highlighting the main study findings. The 29.98% of the bibliography cited in the study belongs to the previous five years.
- Thank you
Overall, it is an interesting study and should be considered for publication in Pharmacy, once the minor revisions proposed have been resolved.
- Thank you

Round 2
Reviewer 1 Report
Comments and Suggestions for Authors
For some reasons, I am not quiet convinced on your response to my original concern on the scale you used. Quality deployment function that you cited as a source of weight assignment is not clear to me whether it is an appropriate tool.
Perhaps you educate/convince me a bit more, so I have a fair assessment of your paper.
No comments
Author Response

(The authors gave the same response as above.)
